# EXPLORATORY NOT EXPLANATORY: COUNTERFACTUAL ANALYSIS OF SALIENCY MAPS FOR DEEP REINFORCEMENT LEARNING

**Akanksha Atrey, Kaleigh Clary & David Jensen**
University of Massachusetts Amherst
{aatrey,kclary,jensen}@cs.umass.edu

## ABSTRACT

Saliency maps are frequently used to support explanations of the behavior of deep reinforcement learning (RL) agents. However, a review of how saliency maps are used in practice indicates that the derived explanations are often unfalsifiable and can be highly subjective. We introduce an empirical approach grounded in counterfactual reasoning to test the hypotheses generated from saliency maps and assess the degree to which they correspond to the semantics of RL environments. We use Atari games, a common benchmark for deep RL, to evaluate three types of saliency maps. Our results show the extent to which existing claims about Atari games can be evaluated and suggest that saliency maps are best viewed as an exploratory tool rather than an explanatory tool.

## 1 INTRODUCTION

Saliency map methods are a popular visualization technique that produce heatmap-like output highlighting the importance of different regions of some visual input. They are frequently used to explain how deep networks classify images in computer vision applications (Simonyan et al., 2014; Zeiler & Fergus, 2014; Springenberg et al., 2015; Ribeiro et al., 2016; Dabkowski & Gal, 2017; Fong & Vedaldi, 2017; Selvaraju et al., 2017; Shrikumar et al., 2017; Smilkov et al., 2017; Zhang et al., 2018) and to explain how agents choose actions in reinforcement learning (RL) applications (Bogdanovic et al., 2015; Wang et al., 2016; Zahavy et al., 2016; Greydanus et al., 2017; Iyer et al., 2018; Sundar, 2018; Yang et al., 2018; Annasamy & Sycara, 2019).

Saliency methods in computer vision and reinforcement learning use similar procedures to generate these maps. However, the temporal and interactive nature of RL systems presents a unique set of opportunities and challenges. Deep models in reinforcement learning select sequential actions whose effects can interact over long time periods. This contrasts strongly with visual classification tasks, in which deep models merely map from images to labels. For RL systems, saliency maps are often used to assess an agent's *internal representations* and *behavior* over multiple frames in the environment, rather than to assess the importance of specific pixels in classifying images.

Despite their common use to explain agent behavior, it is unclear whether saliency maps provide useful explanations of the behavior of deep RL agents. Some prior work has evaluated the applicability of saliency maps for explaining the behavior of image classifiers (Samek et al., 2017; Adebayo et al., 2018; Kindermans et al., 2019), but there is not a corresponding literature evaluating the applicability of saliency maps for explaining RL agent behavior.

In this work, we develop a methodology grounded in counterfactual reasoning to empirically evaluate the explanations generated using saliency maps in deep RL. Specifically, we:

**C1** Survey the ways in which saliency maps have been used as evidence in explanations of deep RL agents.

**C2** Describe a new interventional method to evaluate the inferences made from saliency maps.

**C3** Experimentally evaluate how well the pixel-level inferences of saliency maps correspond to the semantic-level inferences of humans.

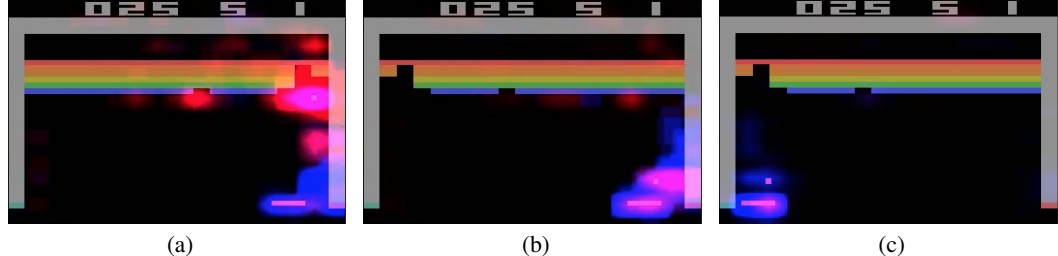

|  (a)  |  (b)  |  (c)  |

Figure 1: Three examples of perturbation saliency maps generated from the same model on different inputs: (a) a frame taken from an episode of agent play in Breakout, (b) the same frame with the brick pattern reflected across the vertical axis, and (c) the original frame with the ball, paddle and brick pattern reflected across the vertical axis. The blue and red regions represent their importance in action selection and reward estimation from the current state, respectively. The pattern and intensity of saliency around the tunnel is not symmetric in either reflection intervention, indicating that a popular hypothesis (agents learn to aim at tunnels) does not hold for all possible tunnels.

## 2 INTERPRETING SALIENCY MAPS IN DEEP RL

Consider the saliency maps generated from a deep RL agent trained to play the Atari game Breakout. The goal of Breakout is to use the paddle to keep the ball in play so it hits bricks, eliminating them from the screen. Figure 1a shows a sample frame with its corresponding saliency. Note the high salience on the missing section of bricks ("tunnel") in Figure 1a.

Creating a tunnel to target bricks at the top layers is one of the most high-profile examples of agent behavior being explained according to *semantic*, human-understandable concepts (Mnih et al., 2015). Given the intensity of saliency on the tunnel in 1a, it may seem reasonable to infer that this saliency map provides evidence that the agent has learned to aim at tunnels. If this is the case, moving the horizontal position of the tunnel should lead to similar saliency patterns on the new tunnel. However, Figures 1b and 1c show that the salience pattern is not preserved. Neither the presence of the tunnel, nor the relative positioning of the ball, paddle, and tunnel, are responsible for the intensity of the saliency observed in Figure 1a.

### 2.1 SALIENCY MAPS AS INTERVENTIONS

Examining how some of the technical details of reinforcement learning interact with saliency maps can help explain both the potential utility and the potential pitfalls of interpreting saliency maps. RL methods enable agents to learn how to act effectively within an environment by repeated interaction with that environment. Certain states in the environment give the agent positive or negative reward. The agent learns a *policy*, a mapping between states and actions according to these reward signals. The goal is to learn a policy that maximizes the discounted sum of rewards received while acting in the environment (Sutton & Barto, 1998). Deep reinforcement learning uses deep neural networks to represent policies. These models enable interaction with environments requiring high-dimensional state inputs (e.g., Atari games).

Consider the graphical model in Figure 2a representing the deep RL system for a vision-based game environment. Saliency maps are produced by performing some kind of intervention $M$ on this system and calculating the difference in logits produced by the original and modified images. The interventions used to calculate saliency for deep RL are performed at the pixel level (red node and arrow in Figure 2a). These interventions change the conditional probability distribution of "Pixels" by giving it another parent (Pearl, 2000).

Functionally, this can be accomplished through a variety of means, including changing the color of the pixel (Simonyan et al., 2014), adding a gray mask (Zeiler & Fergus, 2014), blurring a small region (Greydanus et al., 2017), or masking objects with the background color (Iyer et al., 2018). The interventions $M$ are used to simulate the effect of the absence of the pixel(s) on the network's output. Note however that these interventions change the image in a way that is inconsistent with the

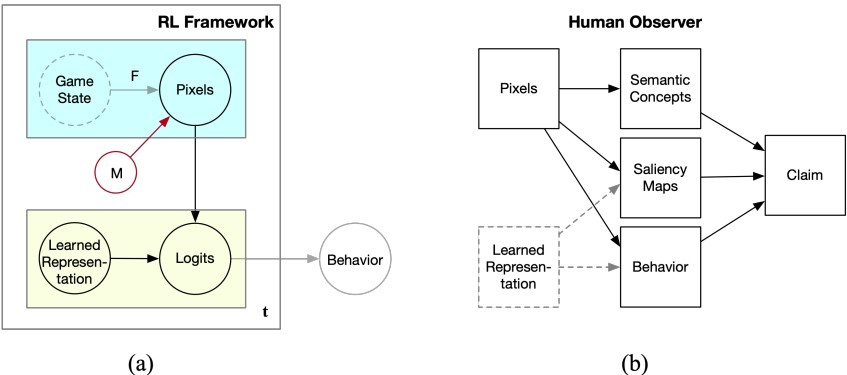

(a)                            (b)

Figure 2: (a) Causal graphical model of the relationships between an RL agent (yellow plate) and an image-based environment (blue plate). The environment maintains some (usually latent) game state. Some function $F$ produces a high-dimensional pixel representation of game state ("Pixels"). The learned network takes this pixel image and produces logits used to select an action. Temporally extended sequences of this action selection procedure result in observed agent behavior. "M" represents interventions made by saliency maps. Such interventions are not naturalistic and are inconsistent with the generative process F; (b) conceptual diagram of how a human observer infers explanations. Hypotheses ("Claims") about the semantic features identified by the learned policy are proposed by reasoning backwards about what representation, often latent, might jointly produce the observed saliency pattern and agent behavior.

generative process $F$. They are not "naturalistic" interventions. This type of intervention produces images for which the learned network function may not be well-defined.

## 2.2 EXPLANATIONS FROM SALIENCY MAPS

To form explanations of agent behavior, human observers combine information from saliency maps, agent behavior, and semantic concepts. Figure 2b shows a system diagram of how these components interact. We note that semantic concepts are often identified visually from the pixel output as the *game state* is typically latent.

Counterfactual reasoning has been identified as a particularly effective way to present explanations of the decision boundaries of deep models (Mittelstadt et al., 2019). Humans use counterfactuals to reason about the enabling conditions of particular outcomes, as well as to identify situations where the outcome would have occurred even in the absence of some action or condition (de Graaf & Malle, 2017; Byrne, 2019). Saliency maps provide a kind of pixel-level counterfactual, but if the goal is to explain agent behavior according to semantic concepts, interventions at the pixel level seem unlikely to be sufficient.

Since many semantic concepts may map to the same set of pixels, it may be difficult to identify the functional relationship between changes in pixels and changes in network output according to semantic concepts or game state (Chalupka et al., 2015). Researchers may be interpreting differences in network outputs as evidence of differences in semantic concepts. However, changes in pixels do not guarantee changes in semantic concepts or game state.

In terms of changes to pixels, semantic concepts, and game state, we distinguish among three classes of interventions: *distortion*, *semantics-preserving*, and *fat-hand* (see Table 1). Semantics-preserving and fat-hand interventions are defined with respect to a specific set of semantic concepts. Fat-hand interventions change game state in such a way that the semantic concepts of interest are also altered.

The pixel-level manipulations used to produce saliency maps primarily result in distortion interventions, though some saliency methods (e.g., object-based) may conceivably produce semantics-preserving or fat-hand interventions as well. Pixel-level interventions are not guaranteed to produce changes in semantic concepts, so counterfactual evaluations that apply semantics-preserving interventions may be a more appropriate approach for precisely testing hypotheses of behavior.

| Intervention | Change in... | | | Examples |
|---|---|---|---|---|
| | pixels | game state | semantic concepts | |
| Distortion | ✓ | ✗ | ✗ | Adversarial ML (Szegedy et al., 2014) |
| Semantics-preserving | ✓ | ✓ | ✗ | Reflection across a line of symmetry |
| Fat-hand | ✓ | ✓ | ✓ | Teleporting the agent to a new position |

Table 1: Categories of interventions on images. *Distortion interventions* change pixels without changing game state or semantic concepts. Pixel perturbations in adversarial ML add adversarial noise to images to change the output of the network without making human-perceptible changes in the image. *Semantics-preserving interventions* are manipulations of game state that result in an image that preserves some semantic concept of interest. Reflections across lines of symmetry typically alter aspects of game state, but do not meaningfully change any semantic information about the scene. *"Fat-hand" interventions* are manipulations intended to measure the effect of some specific treatment, but which unintentionally alter other relevant aspects of the system. The term is drawn from the literature on causal modeling (Scheines, 2005).

# 3   SURVEY OF USAGE OF SALIENCY MAPS IN DEEP RL LITERATURE

To assess how saliency maps are typically used to make inferences regarding agent behavior, we surveyed recent conference papers in deep RL. We focused our pool of papers on those that use saliency maps to generate explanations or make claims regarding agent behavior. Our search criteria consisted of examining papers that cited work that first described any of the following four types of saliency maps:

**Jacobian Saliency.** Wang et al. (2016) extend gradient-based saliency maps to deep RL by computing the Jacobian of the output logits with respect to a stack of input images.

**Perturbation Saliency.** Greydanus et al. (2017) generate saliency maps by perturbing the original input image using a Gaussian blur of the image and measure changes in policy from removing information from a region.

**Object Saliency.** Iyer et al. (2018) use template matching, a common computer vision technique (Brunelli, 2009), to detect (template) objects within an input image and measure salience through changes in Q-values for masked and unmasked objects.

**Attention Saliency.** Most recently, attention-based saliency mapping methods have been proposed to generate interpretable saliency maps (Mott et al., 2019; Nikulin et al., 2019).

From a set of 90 papers, we found 46 claims drawn from 11 papers that cited and used saliency maps as evidence in their explanations of agent behavior. The full set of claims are given in Appendix C.

## 3.1   SURVEY RESULTS

We found three categories of saliency map usage, summarized in Table 2. First, all claims interpret salient areas as a proxy for agent focus. For example, a claim about a Breakout agent notes that the network is focusing on the paddle and little else (Greydanus et al., 2017).

Second, 87% of the claims in our survey propose hypotheses about the features of the learned policy by reasoning backwards about what representation might jointly produce the observed saliency pattern and agent

| | Discuss Focus | Generate Explanation | Evaluate Explanation |
|---|---|---|---|
| Jacobian | 21 | 19 | 0 |
| Perturbation | 11 | 9 | 1 |
| Object | 5 | 4 | 2 |
| Attention | 9 | 8 | 0 |
| Total Claims | 46 | 40 | 3 |

Table 2: Summary of the survey on usage of saliency maps in deep RL. Columns represent categories of saliency map usage, and rows represent categories of saliency map methods, with each cell denoting the number of claims in those categories. Individual claims may be counted in multiple columns.

behavior. These types of claims either develop an *a priori* explanation of behavior and evaluate it using saliency, or they propose an *ad hoc* explanation after observing saliency to reason about how the agent is using salient areas. One *a priori* claim notes that the displayed score is the only differing factor between two states and evaluates that claim by noting that saliency focuses on these pixels (Zahavy et al., 2016). An *ad hoc* claim about a racing game notes that the agent is recognizing a time-of-day cue from the background color and acting to prepare for a new race (Yang et al., 2018).

Finally, only 7% (3 out of 46) of the claims drawn from saliency maps are accompanied by additional or more direct experimental evidence. One of these attempts to corroborate the interpreted saliency behavior by obtaining additional saliency samples from multiple runs of the game. The other two attempt to manipulate semantics in the pixel input to assess the agent's response by, for example, adding an additional object to verify a hypothesis about memorization (Annasamy & Sycara, 2019).

## 3.2 COMMON PITFALLS IN CURRENT USAGE

In the course of the survey, we also observed several more qualitative characteristics of how saliency maps are routinely used.

**Subjectivity.** Recent critiques of machine learning have already noted a worrying tendency to conflate speculation and explanation (Lipton & Steinhardt, 2018). Saliency methods are not designed to formalize an abstract human-understandable concept such as "aiming" in Breakout, and they do not provide a means to quantitatively compare semantically meaningful consequences of agent behavior. This leads to subjectivity in the conclusions drawn from saliency maps.

**Unfalsiability.** One hallmark of a scientific hypothesis or claim is falsifiability (Popper, 1959). If a claim is false, its falsehood should be identifiable from some conceivable experiment or observation. One of the most disconcerting practices identified in the survey is the presentation of unfalsifiable interpretations of saliency map patterns. An example: "A diver is noticed in the saliency map but misunderstood as an enemy and being shot at" (see Appendix C). It is unclear how we might falsify an abstract concept such as "misunderstanding".

**Cognitive Biases.** Current theory and evidence from cognitive science implies that humans learn complex processes, such as video games, by categorizing objects into abstract classes and by inferring causal relationships among instances of those classes (Tenenbaum & Niyogi, 2003; Dubey et al., 2018). Our survey suggests that researchers infer that: (1) salient regions map to learned representations of semantic concepts (e.g., ball, paddle), and (2) the relationships among the salient regions map to high-level behaviors (e.g., tunnel-building, aiming). Researchers' expectations impose a strong bias on both the existence and nature of these mappings.

## 4 METHODOLOGY

Our survey indicates that many researchers use saliency maps as an explanatory tool to infer the representations and processes behind an agent's behavior. However, the extent to which such inferences are valid has not been empirically evaluated under controlled conditions.

In this section, we show how to generate falsifiable hypotheses from saliency maps and propose an intervention-based approach to verify the hypotheses generated from saliency maps. We intervene on game state to produce counterfactual semantic conditions. This provides a concrete methodology to assess the relationship between saliency and learned semantic representations.

**Building Falsifiable Hypotheses from Saliency Maps.** Though saliency maps may not relate directly to semantic concepts, they may still be an effective tool for exploring hypotheses about agent behavior. As we show schematically in Figure 2b, claims or explanations informed by saliency maps have three components: semantic concepts, saliency, and behavior. Recall that our survey indicates that researchers often attempt to infer aspects of the network's learned representations from saliency patterns. Let $X$ be a subset of the semantic concepts that can be inferred from the input image. Let $B$ represent behavior, or aggregate actions, over temporally extended sequences of frames, and let $R$ be a representation that is a function of some pixels that the agent learns during training.

To create scientific claims from saliency maps, we recommend using a relatively standard pattern which facilitates objectivity and falsifiability:

{concept set $X$} is salient $\implies$ agent has learned {representation $R$} resulting in {behavior $B$}.

Consider the Breakout brick reflection example presented in Section 2. The hypothesis introduced ("the agent has learned to aim at tunnels") can be reformulated as: *bricks* are salient $\implies$ agent has learned to *identify a partially complete tunnel* resulting in *maneuvering the paddle to hit the ball toward that region*. Stating hypotheses in this format implies falsifiable claims amenable to empirical analysis.

**Counterfactual Evaluation of Claims.** As indicated in Figure 2, the learned representation and pixel input share a relationship with saliency maps generated over a sequence of frames. Given that the representation learned is static, the relationship between the learned representation and saliency should be invariant under different manipulations of pixel input. We use this property to assess saliency under counterfactual conditions.

We generate counterfactual conditions by intervening on the RL environment. Prior work has focused on manipulating the pixel input. However, this does not modify the underlying latent game state. Instead, we intervene directly on *game state*. In the do-calculus formalism (Pearl, 2000), this shifts the intervention node in Figure 2a to game state, which leaves the generative process $F$ of the pixel image intact.

We employ TOYBOX, a set of fully parameterized implementation of Atari games (Foley et al., 2018), to generate interventional data under counterfactual conditions. The interventions are dependent on the mapping between semantic concepts and learned representations in the hypotheses. Given a mapping between concept set $X$ and a learned representation $R$, any intervention would require meaningfully manipulating the state in which $X$ resides to assess the saliency on $X$ under the semantic treatment applied. Saliency on $x \in X$ is defined as the average saliency over a bounding-box[1] around $x$.

Since the learned policies should be semantically invariant under manipulations of the RL environment, by intervening on state, we can verify whether the counterfactual states produce expected patterns of saliency on the associated concept set $X$. If the counterfactual saliency maps reflect similar saliency patterns, this provides stronger evidence that the observed saliency indicates the agent has learned representation R corresponding to semantic concept set $X$.

## 5 EVALUATION OF HYPOTHESES ON AGENT BEHAVIOR

We conduct three case studies to evaluate hypotheses about the relationship between semantic concepts and semantic processes formed from saliency maps. Each case study uses observed saliency maps to identify hypotheses in the format described in Section 4. The hypotheses were generated by watching multiple episodes and noting atypical, interesting or popular behaviors from saliency maps. In each case study, we produce Jacobian, perturbation and object saliency maps from the same set of counterfactual states. We include examples of each map in Appendix A. Using TOYBOX allows us to produce counterfactual states and to generate saliency maps in these altered states. The case studies are conducted on two Atari games, Breakout and Amidar.[2] The deterministic nature of both games allows some stability in the way we interpret the network's action selection. Each map is produced from an agent trained with A2C (Mnih et al., 2016) using a CNN-based (Mnih et al., 2015) OpenAI Baselines implementation (Dhariwal et al., 2017) with default hyperparameters (see Appendix B for more details). Our choice of model is arbitrary. The emphasis of this work is on *methods* of explanation, not the explanations themselves.

---

[1]TOYBOX provides transparency into the position of each object (measured at the center of sprite), which we use as the center of the bounding boxes in our experiments.

[2]The code pertaining to the experiments can be found at `https://github.com/KDL-umass/saliency_maps`.

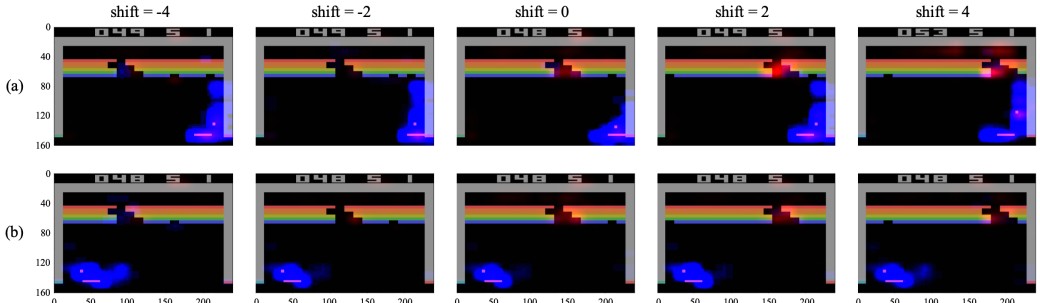

Figure 3: Interventions on brick configurations in Breakout. (a) saliency after shifting the brick positions by some pixels where shift=0 represents the original frame; (b) saliency after shifting the brick positions, ball, and paddle to the left. The pattern and intensity of saliency around the tunnel is not symmetric in the reflection interventions.

**Case Study 1: Breakout Brick Translation.**    Here we evaluate the behavior from Section 2:

> **Hypothesis 1**: {bricks} are salient $\implies$ agent has learned to {identify a partially complete tunnel} resulting in {maneuvering the paddle to hit the ball toward that region}.

To evaluate this hypothesis, we intervene on the state by translating the brick configurations horizontally. Because the semantic concepts relating to the tunnel are preserved under translation, we expect salience will be nearly invariant to the horizontal translation of the brick configuration. Figure 3a depicts saliency after intervention. Salience on the tunnel is less pronounced under left translation, and more pronounced under right translation. Since the paddle appears on the right, we additionally move the ball and paddle to the far left (Figure 3b).

*Conclusion.* Temporal association (e.g. formation of a tunnel followed by higher saliency) does not generally imply causal dependence. In this case, tunnel formation and salience appear to be confounded by location or, at least, the dependence of these phenomena are highly dependent on location.

**Case Study 2: Amidar Score.**    Amidar is a Pac-Man-like game in which an agent attempts to completely traverse a series of passages while avoiding enemies. The yellow sprite that indicates the location of the agent is almost always salient in Amidar. Surprisingly, the displayed score is often as salient as the yellow sprite throughout the episode with varying levels of intensity. This can lead to multiple hypotheses about the agent's learned representation: (1) the agent has learned to associate increasing score with higher reward; (2) due to the deterministic nature of Amidar, the agent has created a lookup table that associates its score and its actions. We can summarize these as follows:

> **Hypothesis 2**: {score} is salient $\implies$ agent has learned to {use score as a guide to traverse the board} resulting in {successfully following similar paths in games}.

To evaluate hypothesis 2, we designed four interventions on score:

- *intermittent_reset*: modify the score to 0 every $x \in [5, 20]$ timesteps.
- *random_varying*: modify the score to a random number between [1,200] every $x \in [5, 20]$ timesteps.
- *fixed*: select a score from [0,200] and fix it for the whole game.
- *decremented*: modify score to be 3000 initially and decrement score by $d \in [1, 20]$ at every timestep.

Figures 4a and 4b show the result of intervening on displayed score on reward and saliency intensity, measured as the average saliency over a 25x15 bounding box, respectively for the first 1000 timesteps of an episode. The mean is calculated over 50 samples. If an agent died before 1000 timesteps, the last reward was extended for the remainder of the timesteps and saliency was set to zero.

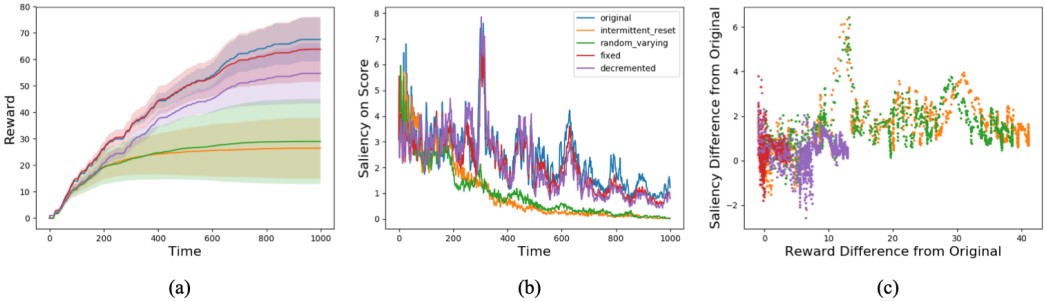

Figure 4: Interventions on displayed score in Amidar. The legend in (b) applies to all figures. (a) reward over time for different interventions on displayed score; (b) object saliency on displayed score over time; (c) correlation between the differences in reward and object saliency from the original trajectory. Interventions on displayed score result in differing levels of degraded performance but produce similar saliency maps, suggesting that agent behavior as measured by rewards is underdetermined by salience.

Using reward as a summary of agent behavior, different interventions on score produce different agent behavior. Total accumulated reward differs over time for all interventions, typically due to early agent death. However, salience intensity patterns of all interventions follow the original trajectory very closely. Different interventions on displayed score cause differing degrees of degraded performance (Figure 4a) despite producing similar saliency maps (Figure 4b), indicating that agent behavior is underdetermined by salience. Specifically, the salience intensity patterns are similar for the *control*, *fixed*, and *decremented* scores, while the non-ordered score interventions result in degraded performance. Figure 4c indicates only very weak correlations between the difference-in-reward and difference-in-saliency-under-intervention as compared to the original trajectory. Correlation coefficients range from 0.041 to 0.274, yielding insignificant p-values for all but one intervention. See full results in Appendix E.1, Table 6.

Similar trends are noted for Jacobian and perturbation saliency methods in Appendix E.1.

*Conclusion.* The existence of a high correlation between two processes (e.g., incrementing score and persistence of saliency) does not imply causation. Interventions can be useful in identifying the common cause leading to the high correlation.

**Case Study 3: Amidar Enemy Distance.** Enemies are salient in Amidar at varying times. From visual inspection, we observe that enemies close to the player tend to have higher saliency. Accordingly, we generate the following hypothesis:

> **Hypothesis 3**: {enemy} is salient $\implies$ agent has learned to {identify enemies close to it} resulting in {successful avoidance of enemy collision}.

Without directly intervening on the game state, we can first identify whether the player-enemy distance and enemy saliency is correlated using observational data. We collect 1000 frames of an episode of Amidar and record the Manhattan distance between the midpoints of the player and enemies, represented by 7x7 bounding boxes, along with the object salience of each enemy. Figure 5a shows the distance of each enemy to the player over time with saliency intensity represented by the shaded region. Figure 5b shows the correlation between the distance to each enemy and the corresponding saliency. Correlation coefficients and significance values are reported in Table 3. It is clear that there is no correlation between saliency and distance of each enemy to the player.

Given that statistical dependence is almost always a necessary pre-condition for causation, we expect that there will not be any causal dependence. To further examine this, we intervene on enemy positions of salient enemies at each timestep by moving the enemy closer and farther away from the player. Figure 5c contains these results. Given Hypothesis 3, we would expect to see an increasing trend in saliency for enemies closer to the player. However, the size of the effect is close to 0 (see Table 3). In addition, we find no correlation in the enemy distance experiments for the Jacobian or perturbation saliency methods (included in Appendix E.2).

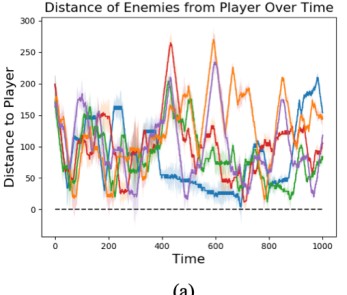 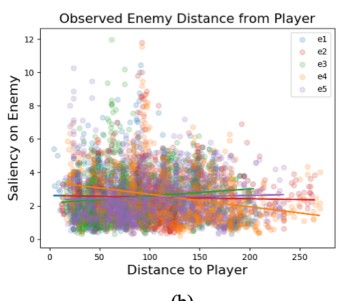 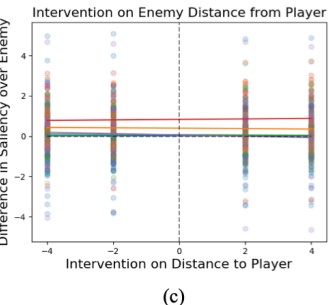

| (a) | (b) | (c) |

Figure 5: Interventions on enemy location in Amidar. The legend in (b) applies to all figures. (a) the distance-to-player of each enemy in Amidar, observed over time, where saliency intensity is represented by the shaded region around each line; (b) the distance-to-player and saliency, with linear regressions, observed for each enemy; (c) variation in enemy saliency when enemy position is varied by intervention. The plots suggest that there is no substantial correlation and no causal dependence between distance-to-player and object saliency.

|  | **Observational** | | | **Interventional** | | |
| Enemy | slope | $p$-value | $r$ | slope | $p$-value | $r$ |
|---|---|---|---|---|---|---|
| 1 | -0.001 | 0.26 | -0.036 | -0.001 | 0.97 | -0.001 |
| 2 | -0.001 | 0.35 | -0.298 | 0.013 | 0.68 | 0.039 |
| 3 | 0.004 | 1.59e−4 | 0.119 | -0.008 | 0.79 | -0.022 |
| 4 | -0.008 | 8.26e−19 | -0.275 | -0.011 | 0.75 | -0.028 |
| 5 | 0.001 | 0.13 | 0.047 | -0.033 | 0.47 | -0.063 |

Table 3: Numeric results from regression analysis for the observational and interventional results in Figures 5b and c. The results indicate a very small strength of effect (slope) for both observational and interventional data and a small correlation coefficient ($r$), suggesting that there is, at best, only a very weak causal dependence of saliency on distance-to-player.

*Conclusion.* Spurious correlations, or misinterpretations of existing correlation, can occur between two processes (e.g. correlation between player-enemy distance and saliency), and human observers are susceptible to identifying spurious correlations (Simon, 1954). Spurious correlations can sometimes be identified from observational analysis without requiring interventional analysis.

## 6  DISCUSSION AND RELATED WORK

Thinking counterfactually about the explanations generated from saliency maps facilitates empirical evaluation of those explanations. The experiments above show some of the difficulties in drawing conclusions from saliency maps. These include the tendency of human observers to incorrectly infer association between observed processes, the potential for experimental evidence to contradict seemingly obvious observational conclusions, and the challenges of potential confounding in temporal processes.

One of the main conclusions from this evaluation is that *saliency maps are an exploratory tool rather than an explanatory tool* for evaluating agent behavior in deep RL. Saliency maps alone cannot be reliably used to infer explanations and instead require other supporting tools. This can include combining evidence from saliency maps with other explanation methods or employing a more experimental approach to evaluation of saliency maps such as the approach demonstrated in the case studies above.

The framework for generating falsifiable hypotheses suggested in Section 4 can assist with designing more specific and falsifiable explanations. The distinction between the components of an explanation, particularly the semantic concept set $X$, learned representation $R$ and observed behavior $B$, can further assist in experimental evaluation. Note that the semantic space devised by an agent might be quite different from the semantic space given by the latent factors of the environment. It is cru-

cial to note that this mismatch is one aspect of what plays out when researchers create hypotheses about agent behavior, and the methodology we provide in this work demonstrates how to evaluate hypotheses that reflect that mismatch.

**Generalization of Proposed Methodology.** The methodology presented in this work can be easily extended to other vision-based domains in deep RL. Particularly, the framework of the graphical model introduced in Figure 2a applies to all domains where the input to the network is image data. An extended version of the model for Breakout can be found in Appendix 7.

We propose intervention-based experimentation as a primary tool to evaluate the hypotheses generated from saliency maps. Yet, alternative methods can identify a false hypothesis even earlier. For instance, evaluating statistical dependence alone can provide strong evidence against causal dependence (e.g., Case Study 3). In this work, we employ a particularly capable simulation environment (TOYBOX). However, limited forms of evaluation may be possible in non-intervenable environments, though they may be more tedious to implement. For instance, each of the interventions conducted in Case Study 1 can be produced in an observation-only environment by manipulating the pixel input (Brunelli, 2009; Chalupka et al., 2015). Developing more experimental systems for evaluating explanations is an open area of research.

This work analyzes explanations generated from feed-forward deep RL agents. However, the proposed methodology is not model dependent, and aspects of the approach will carry over to recurrent deep RL agents. The proposed methodology would not work well for repeated interventions on recurrent deep RL agents due to their capacity for memorization.

**Explanations in Deep RL.** Prior work has introduced alternatives to the use of saliency maps to support explanation of deep RL agents. Some of these methods also use counterfactual reasoning to develop explanations. TOYBOX was developed to support experimental evaluation and behavioral tests of deep RL models (Tosch et al., 2019). Olson et al. (2019) use a generative deep learning architecture to produce counterfactual states resulting in the agent taking a different action. Others have proposed alternative methods for developing semantically meaningful interpretations of agent behavior. Juozapaitis et al. (2019) use reward decomposition to attribute policy behaviors according to semantically meaningful components of reward. Verma et al. (2018) use domain-specific languages for policy representation, allowing for human-readable policy descriptions.

**Evaluation and Critiques of Saliency Maps.** Prior work in the deep network literature has evaluated and critiqued saliency maps. Kindermans et al. (2019) and Adebayo et al. (2018) demonstrate the utility of saliency maps by adding random variance in input. Seo et al. (2018) provide a theoretical justification of saliency and hypothesize that there exists a correlation between gradients-based saliency methods and model interpretation. Samek et al. (2017) and Hooker et al. (2019) present evaluations of existing saliency methods for image classification.

## 7   CONCLUSIONS

We conduct a survey of uses of saliency maps, propose a methodology to evaluate saliency maps, and examine the extent to which the agent's learned representations can be inferred from saliency maps. We investigate how well the pixel-level inferences of saliency maps correspond to the semantic concept-level inferences of human-level interventions. Our results show saliency maps cannot be trusted to reflect causal relationships between semantic concepts and agent behavior. We recommend saliency maps to be used as an exploratory tool, not explanatory tool.

### ACKNOWLEDGMENTS

Thanks to Emma Tosch, Amanda Gentzel, Deep Chakraborty, Abhinav Bhatia, Blossom Metevier, Chris Nota, Karthikeyan Shanmugam and the anonymous ICLR reviewers for thoughtful comments and contributions. This material is based upon work supported by the United States Air Force under Contract No, FA8750-17-C-0120. Any opinions, findings and conclusions or recommendations expressed in this material are those of the author(s) and do not necessarily reflect the views of the United States Air Force.

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

## APPENDICES

## A SALIENCY METHODS

Figure 6 shows example saliency maps of the three saliency methods evaluated in this work, namely perturbation, object and Jacobian, for Amidar.

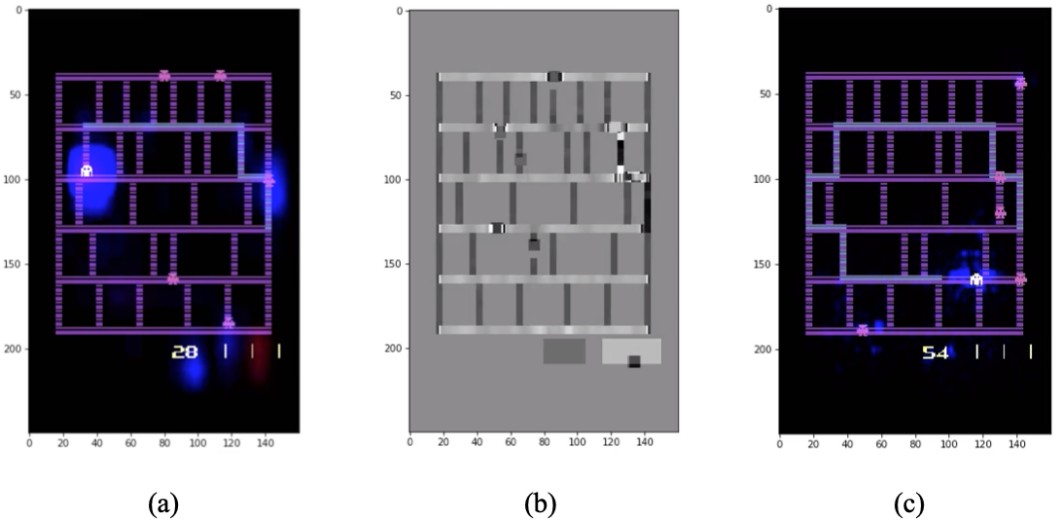

| (a) | (b) | (c) |

Figure 6: Examples of (a) perturbation saliency method (Greydanus et al., 2017), (b) object saliency method (Iyer et al., 2018), and (c) Jacobian saliency method (Wang et al., 2016), for Amidar.

## B MODEL

We use the OpenAI Baselines' implementation (Dhariwal et al., 2017) of an A2C model (Mnih et al., 2016) to train the RL agents on Breakout and Amidar. The model uses the CNN architecture proposed by Mnih et al. (2015). Each agent is trained for 40 million iterations using RMSProp with default hyperparameters (Table 4).

| Hyperparameter | Value |
| --- | --- |
| Learning rate | 7e−4 |
| Learning rate schedule | Linear |
| # Iterations | 40,000,000 |
| Value function coefficient | 0.5 |
| Policy function coefficient | 0.01 |
| RMSProp epsilon | 1e−5 |
| RMSProp decay | 0.99 |
| Reward discounting parameter | 0.99 |
| Max gradient (clip) | 0.5 |

Table 4: Hyperparameters used in training the A2C model.

## C  SURVEY OF USAGE OF SALIENCY MAPS IN DEEP RL LITERATURE

We conducted a survey of recent literature to assess how saliency maps are used to interpret agent behavior in deep RL. We began our search by focusing on work citing the following four types of saliency maps: Jacobian (Wang et al., 2016), perturbation (Greydanus et al., 2017), object (Iyer et al., 2018) and attention (Mott et al., 2019). Papers were selected if they employed saliency maps to create explanations regarding agent behavior. This resulted in selecting 46 claims from 11 papers. These 11 papers have appeared at ICML (3), NeurIPS (1), AAAI (2), ArXiv (3), OpenReview (1) and as a thesis (1). There are several model-specific saliency mapping methods that we excluded from our survey.

Following is the full set of claims. All claims are for Atari games. The *Reason* column represents whether an explanation for agent behavior was provided (Y/N) and the *Exp* column represents whether an experiment was conducted to evaluate the explanation (Y/N).

| Claim | Game | Saliency Type | Reason | Exp |
|---|---|---|---|---|
| Greydanus et al. (2017) | | | | |
| "The agent is positioning its own paddle, which allows it to return the ball at a specific angle." | Pong | Perturbation | Y | N |
| "Interestingly, from the saliency we see that the agent attends to very little besides its own paddle: not even the ball." | Pong | Perturbation | N | N |
| "After the agent has executed the kill shot, we see that saliency centers entirely around the ball. This makes sense since at this point neither paddle can alter the outcome and their positions are irrelevant." | Pong | Perturbation | Y | N |
| "It appears that the deep RL agent is exploiting the deterministic nature of the Pong environment. It has learned that it can obtain a reward with high certainty upon executing a precise series of actions. This insight...gives evidence that the agent is not robust and has overfit to the particular opponent." | Pong | Perturbation | Y | N |
| "[The agent] had learned a sophisticated aiming strategy during which first the actor and then the critic would 'track' a target. Aiming begins when the actor highlights a particular alien in blue...Aiming ends with the agent shooting at the new target." | Space-Invaders | Perturbation | Y | N |
| "The critic highlights the target in anticipation of an upcoming reward." | Space-Invaders | Perturbation | Y | N |
| "Notice that both actor and critic tend to monitor the area above the ship. This may be useful for determining whether the ship is protected from enemy fire or has a clear shot at enemies." | Space-Invaders | Perturbation | Y | N |
| "We found that the agent enters and exits a 'tunneling mode' over the course of a single frame. Once the tunneling location becomes salient, it remains so until the tunnel is finished." | Breakout | Perturbation | N | N |
| Zahavy et al. (2016) | | | | |
| "A diver is noticed in the saliency map but misunderstood as an enemy and being shot at." | Seaquest | Jacobian | Y | N |

| | | | | |
|---|---|---|---|---|
| "Once the agent finished the first screen it is presented with another one, distinguished only by the score that was accumulated in the first screen. Therefore, an agent might encounter problems with generalizing to the new screen if it over-fits the score pixels. Figure 15 shows the saliency maps of different games supporting our claim that DQN is basing its estimates using these pixels. We suggest to further investigate this, for example, we suggest to train an agent that does not receive those pixels as input." | Breakout | Jacobian | N | N |
| Bogdanovic et al. (2015) | | | | |
| "The line of cars in the upper right are far away and the agent correctly ignores them in favour of focusing on the much more dangerous cars in the lower left." | Freeway | Jacobian | Y | N |
| Yang et al. (2018) | | | | |
| "Firstly, the agent ignores irrelevant features from mountains, sky, the mileage board and empty grounds, and relies on information from the race track to make decisions. Specifically, the agent keeps separate two categories of objects on the race track, i.e., cars and the player." | Enduro | Binary Jacobian | Y | N |
| "On the one hand, the agent locates the player and a local area around it for avoiding immediate collisions with cars. On the other hand, the agent locates the next potential collision targets at different locations, particularly the remote ones." | Enduro | Binary Jacobian | Y | N |
| "Near the completion of the current goal, the agent celebrates in advance. As shown from Fig. 4(d) to Fig. 4(f), the left gaze loses its focus on cars and diverts to the mileage board starting when only 13 cars remain before completion. The previous car tracker now picks up on the important information that it is close to victory, and fixates on the countdown." | Enduro | Binary Jacobian | Y | N |
| "Upon reaching the target, the agent does not receive reward signals until the next day starts. During this period the agent learns to output no-op actions, corresponding to not playing the game." | Enduro | Binary Jacobian | Y | N |
| "When slacking happens, the agent considers the flag signs as important and the road not. The complete reverse in focus as compared to the normal case explains this shift in policy. The flags outweigh the road in importance, since they are signs of absolute zero return." | Enduro | Binary Jacobian | Y | N |
| "(Prepping) As it turns out, the agent recognizes that the time is dawn (right before morning when race starts) from the unique colours of the light gray sky and orange mountains, therefore the agent gets ready early for a head start in the new race." | Enduro | Binary Jacobian | Y | N |
| "When smog partially blocks the forward view, the left gaze loses its focus on cars. It strays off the road into some empty area." | Enduro | Binary Jacobian | Y | N |

| | | | | |
|---|---|---|---|---|
| "In Fig. 5(a), the left gaze detects the two ghosts on the upper-right corner of the map. Therefore, ms pacman, as located by the right gaze, stays in the mid-left section to safely collect dense rewards. " | Pacman | Binary Jacobian | Y | N |
| "In Fig. 5(b), the left gaze locks in on all three vulnerable ghosts in the mid-right section, as ms pacman chases after them." | Pacman | Binary Jacobian | Y | N |
| "In Fig. 5(c), the left gaze detects a newly-appeared cherry at the lower-left warp tunnel entrance. Ms pacman immediately enters the closest opposite tunnel entrance in the shortest path to the cherry." | Pacman | Binary Jacobian | Y | N |
| "In Fig. 5(d), the right gaze locates ms pacman entering the upper-right tunnel. In this case, the left gaze no longer detects a moving object, but predicts the upper-left tunnel as the exiting point." | Pacman | Binary Jacobian | Y | N |
| "As shown in Fig. 5(g), the left gaze locates the last pellet when ms pacman is in the mid-section of the maze. Therefore ms pacman moves towards the pellet." | Pacman | Binary Jacobian | Y | N |
| "In Fig. 5(h), a red ghost appears in the left gaze close to the pellet, causing ms pacman to deviate to the right." | Pacman | Binary Jacobian | Y | N |
| "After changing course, the ghosts approach ms pacman from all directions as shown in Fig. 5(i). Even though the agent detects all the ghosts (they appear in the gazes), ms pacman has no route to escape." | Pacman | Binary Jacobian | Y | N |
| "The left gaze often focuses on white ice blocks that are the destinations of jumping." | Frostbite | Binary Jacobian | N | N |
| "Fig. 6(b) shows the sub-task of the player entering the igloo, after jumping over white ice blocks for building it. The player must avoid the bear when running for the igloo." | Frostbite | Binary Jacobian | Y | N |
| "As it shows, the igloo is two jumps away from completion, and the left gaze focuses on the igloo in advance for preparing to enter." | Frostbite | Binary Jacobian | Y | N |
| Annasamy & Sycara (2019) | | | | |
| "For example, in MsPacman, since visualizations suggest that the agent may be memorizing pacmans positions (also maybe ghosts and other objects), we simply add an extra pellet adjacent to a trajectory seen during training (Figure 7a). The agent does not clear the additional pellet and simply continues to execute actions performed during training (Figure 7b)." | Pacman | Object | Y | Y |
| "Similarly, in case of SpaceInvaders, the agent has a strong bias towards shooting from the leftmost-end (seen in Figure 4). This helps in clearing the triangle like shape and moving to the next level (Figure 7d). However, when triangular positions of spaceships are inverted, the agent repeats the same strategy of trying to shoot from left and fails to clear ships (Figure 7c)." | Space-Invaders | Object | Y | Y |

| | | | | |
|---|---|---|---|---|
| **Goel et al. (2018)** | | | | |
| "In our Breakout results, the network learns to split the paddle into a left and right side, and does not move the middle portion." | Breakout | Object | Y | N |
| "On Beam Rider, our network achieves low loss after learning to segment the games light beams, however these beams are purely visual effects and are unimportant for action selection." | Beam Rider | Object | Y | N |
| "Consequently, the game enemies, which are much smaller in size, are ignored by the network, and the resulting learned representation is not that useful for a reinforcement learning agent." | Beam Rider | Object | N | N |
| **van der Wal et al. (2018)** | | | | |
| "There is a very notable difference between the policy saliency between the two models, where the former one only pays limited attention to the road and almost no attention to the engine indicator, the opposite from fA3C-LSTM. Explicitly,it means masking any regions from the input does not cause much perturbation to the policy when trained with continuous space as targets, likely because the real consequence from a small change in action, e.g. no braking (a3= 0) versus braking (a3= 0.3), can be very substantial but numerically too subtle for the network to capture during optimization on the continuous spectrum." | CarRacing | Perturbation | Y | N |
| **Rupprecht et al. (2018)** | | | | |
| "Analyzing the visualizations on Seaquest, we make an interesting observation. When maximizing the Q-value for the actions, in many samples we see a low or very low oxygen meter. In these cases the submarine would need to ascend to the surface to avoid suffocation. Although the up action is the only sensible choice in this case, we also obtain visualized low oxygen states for all other actions. This implies that the agent has not understood the importance of resurfacing when the oxygen is low. We then run several roll outs of the agent and see that the major cause of death is indeed suffocation and not collision with enemies." | SeaQuest | Perturbation | Y | Y |
| **Mott et al. (2019)** | | | | |
| "The most dominant pattern we observe is that the model learns to attend to task-relevant things in the scene. In most ATARI games that usually means that the player is one of the foci of attention, as well as enemies, power-ups and the score itself (which is an important factor in the calculating the value function)." | SeaQuest | Attention | N | N |

| | | | | |
|---|---|---|---|---|
| "Figure 4 shows a examples of this in Ms Pac-man and Alien  in the both games the model scans through possible paths, making sure there are no enemies or ghosts ahead.  We observe that when it does see a ghost, another path is produced or executed in order to avoid it." | Pacman | Attention | Y | N |
| "In many games we observe that the agent learns to place trip-wires at strategic points in space such that if a game object crosses them a specific action is taken. For example, in Space Invaders two such trip wires are following the player ship on both sides such that if a bullet crosses one of them the agent immediately evades them by moving towards the opposite direction." | Space Invaders | Attention | Y | N |
| "Another example is Breakout where we can see it working in two stages. First the attention is spread out around the general area of the ball, then focuses into a localized line.  Once the ball crosses that line the agent moves towards the ball." | Breakout | Attention | Y | N |
| "As can be seen, the system uses the two modes to make its decisions, some of the heads are content specific looking for opponent cars.  Some are mixed, scanning the horizon for incoming cars and when found, tracking them, and some are location based queries, scanning the area right in front of the player for anything the crosses its path (a trip-wire which moves with the player)." | Enduro | Attention | Y | N |
| "Comparing the attention agent to the baseline agent, we see that the attention agent is sensitive to more focused areas along the possible future trajectory. The baseline agent is more focused on the area immediately in front of the player (for the policy saliency) and on the score, while the attention agent focuses more specifically on the path the agent will follow (for the policy) and on possible future longer term paths (for the value)." | Pacman | Perturbation | Y | N |

| | | | | |
|---|---|---|---|---|
| Nikulin et al. (2019) | | | | |
| "Figs. 4(a)-(b) show Dense FLS digging a tunnel through blocks in Breakout.  The model focuses its attention on the end of the tunnel as soon as it is complete, suggesting that it sees shooting the ball through the tunnel as a good strategy." | Breakout | Attention | Y | N |
| "Figs. 4(c)-(d) depict the same concept of tunneling per-formed by theSparse FLSmodel. Note how it focuses attention on the upper part of the screen after destroying multiple bricks from the top. This attention does not go away after the ball moves elsewhere (not shown in the images).We speculate that this is how the agent models tunneling:rather than having a high-level concept of digging a tunnel,it simply strikes wherever it has managed to strike already." | Breakout | Attention | Y | N |

| | | | |
|---|---|---|---|
| "Figs. 4(e)-(f) illustrate how the Dense FLS model playing Seaquest has learned to attend to in-game objects and, importantly, the oxygen bar at the bottom of the screen. As the oxygen bar is nearing depletion, attention focuses around it,and the submarine reacts by rising to refill its air supply." | Seaquest | Attention | Y | N |
| "Figs. 4(g)-(h) are two consecutive frames where an agent detects a target appearing from the left side of the screen.The bottom part of the screenshots shows how attention in the bottom left corner lights up as soon as a tiny part of the target, only a few pixels wide, appears from the left edge of the screen. In the next frame, the agent will turn left and shoot the target (not shown here). However, the agent completely ignores targets in the top part of the screen, and its attention does not move as they move (also not shown)." | Breakout | Attention | Y | N |

Wang et al. (2016)

| | | | |
|---|---|---|---|
| "The value stream learns to pay attention to the road. The advantage stream learns to pay attention only when there are cars immediately in front, so as to avoid collisions." | Enduro | Jacobian | Y | N |

## D  GENERALIZATION OF CAUSAL GRAPHICAL MODEL

The causal graphical model in Figure 2a can be extended to different domains in RL where the input to the model is an image. This requires extending *game state* to include the underlying *state* variables. Figure 7 shows the extension for Breakout using an A2C model. Note, *game state* includes the underlying state variables for Breakout and *logits* were split into *action logits* and *value* as outputted by A2C.

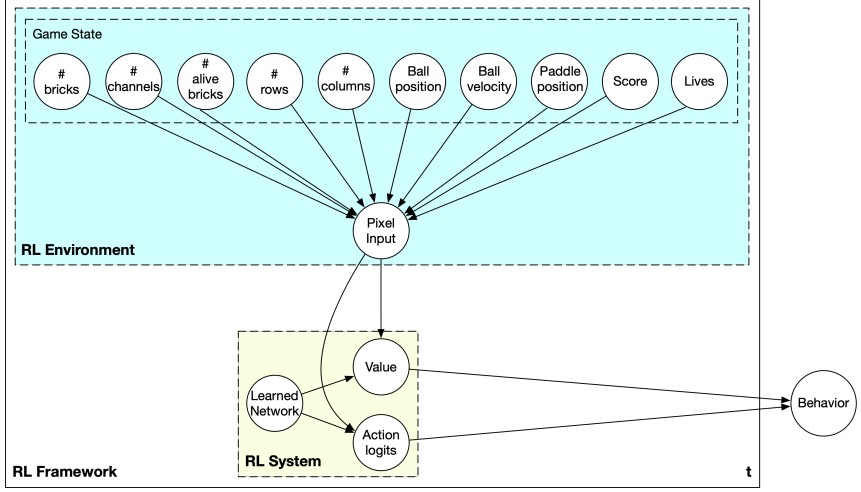

Figure 7: Causal graphical model for Breakout.

|  | Object | | Perturbation | | Jacobian | |
|---|---|---|---|---|---|---|
| Intervention | $r$ | $p$ | $r$ | $p$ | $r$ | $p$ |
| intermittent_reset | 0.274 | 1.09e−18 | -0.10 | 1.52e−3 | 0.149 | 2.26e−6 |
| random_varying | 0.142 | 6.95e−6 | -0.02 | 0.49 | 0.281 | 1.21e−19 |
| fixed | 0.041 | 0.20 | -0.02 | 0.44 | 0.286 | 2.51e−20 |
| decremented | 0.119 | 0.15e−3 | -0.09 | 6.73e−3 | 0.261 | 5.31e−17 |

Table 6: Numeric results representing Pearson's correlation coefficient and $p$-value for the differences in reward and saliency (object, perturbation and Jacobian) from the original trajectory for each intervention. Results show small correlation coefficients ($r$) suggesting that there is a weak relationship between the differences in reward and saliency for the interventions.

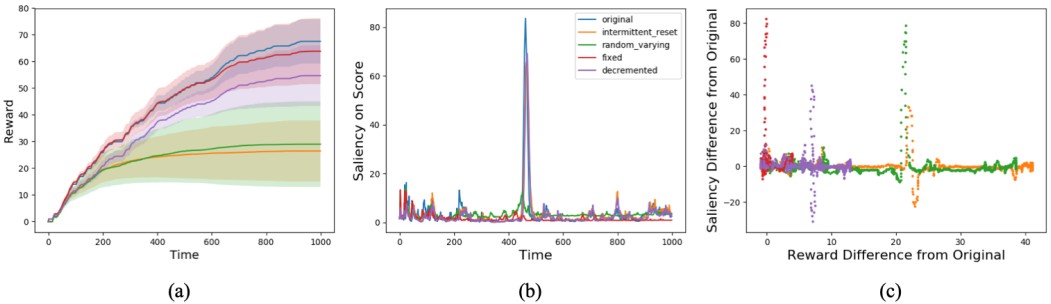

(a)  (b)  (c)

Figure 8: Interventions on displayed score in Amidar (legend in (b) applies to all figures). (a) reward over time for different interventions on displayed score; (b) perturbation saliency on displayed score over time; (c) correlation between the differences in reward and perturbation saliency from the original trajectory. Interventions on displayed score result in differing levels of degraded performance but produce similar saliency maps, suggesting that agent behavior as measured by rewards is underdetermined by salience.

# E   EVALUATION OF HYPOTHESES ON AGENT BEHAVIOR

## E.1   CASE STUDY 2: AMIDAR SCORE

We further evaluated the effects of interventions on displayed score (Section 5) in Amidar on perturbation and Jacobian saliency maps. These results are presented in Figures 8 and 9, respectively.

We also evaluated Pearson's correlation between the differences in reward and saliency with the original trajectory for all three methods (see Table 6). The results support the correlation plots in Figures 4c, 8c and 9c.

## E.2   CASE STUDY 3: AMIDAR ENEMY DISTANCE

We further evaluated the relationship between player-enemy distance and saliency (Section 5) in Amidar on perturbation and Jacobian saliency maps. These results are presented in Figures 10 and 11, respectively. Jacobian saliency performed the worst for the intervention-based experiment, suggesting that there is no impact of player-enemy distance on saliency.

Regression analysis between distance and perturbation and Jacobian saliency can be found in Tables 7 and 8 respectively. The results support the lack of correlation between the observational and interventional distributions. Note, enemy 1 is more salient throughout the game compared to the other four enemies resulting in a larger interventional sample size.

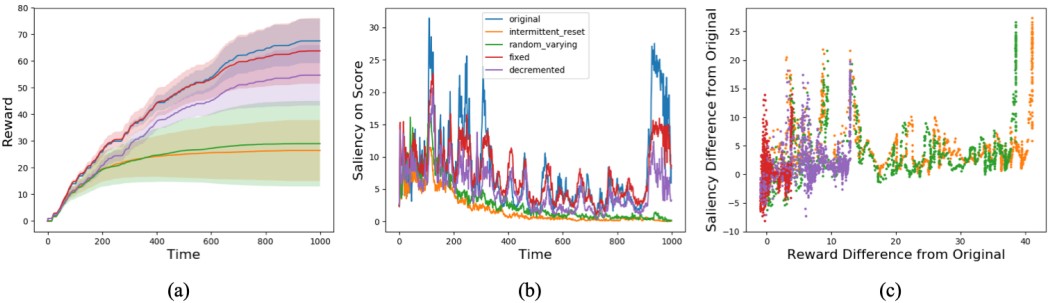

(a)                                      (b)                                      (c)

Figure 9: Interventions on displayed score in Amidar for Jacobian saliency (legend in (b) applies to all figures). (a) reward over time for different interventions on displayed score; (b) saliency on displayed score over time; (c) correlation between the differences in reward and saliency from the original trajectory. Interventions on displayed score result in differing levels of degraded performance but produce similar saliency maps, suggesting that agent behavior as measured by rewards is underdetermined by salience.

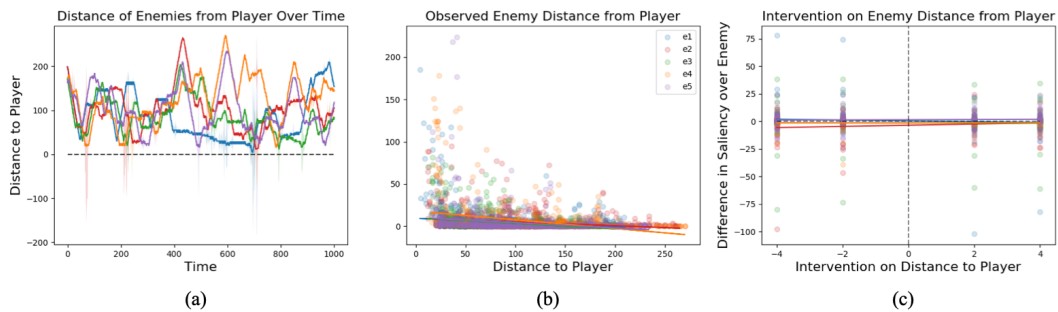

(a)                                      (b)                                      (c)

Figure 10: Interventions on enemy location in Amidar (legend in (b) applies to all figures). (a) the distance-to-player of each enemy in Amidar, observed over time, where saliency intensity is represented by the shaded region around each line; (b) the distance-to-player and saliency, with linear regressions, observed for each enemy; (c) variation in enemy saliency when enemy position is varied by intervention. The plots suggest that there is no substantial correlation and no causal dependence between distance-to-player and perturbation saliency.

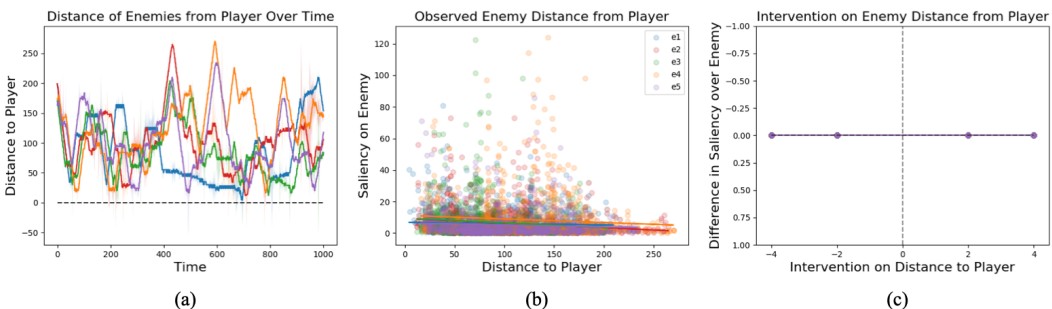

(a)                                      (b)                                      (c)

Figure 11: Interventions on enemy location in Amidar (legend in (b) applies to all figures). (a) the distance-to-player of each enemy in Amidar, observed over time, where saliency intensity is represented by the shaded region around each line; (b) the distance-to-player and saliency, with linear regressions, observed for each enemy; (c) variation in enemy saliency when enemy position is varied by intervention. The plots suggest that there is no substantial correlation and no causal dependence between distance-to-player and Jacobian saliency.

| Enemy | Observational | | | Interventional | | |
|---|---|---|---|---|---|---|
| | Slope | $p$-value | $r$ | Slope | $p$-value | $r$ |
| 1 | -0.052 | 7.04e−10 | -0.204 | -0.374 | 0.01 | -0.126 |
| 2 | -0.043 | 1.09e−9 | -0.191 | 0.536 | 0.16 | 0.132 |
| 3 | -0.065 | 6.04e−20 | -0.283 | -0.039 | 0.92 | -0.008 |
| 4 | -0.103 | 2.57e−25 | -0.320 | -0.005 | 0.98 | 0.003 |
| 5 | -0.062 | 3.44e−12 | -0.217 | -0.083 | 0.71 | 0.032 |

Table 7: Numeric results from regression analysis for the observational and interventional results in Figures 10b and c. The results indicate a very small strength of effect (slope) for both observational and interventional data and a small correlation coefficient ($r$), suggesting that there is, at best, only a very weak causal dependence of saliency on distance-to-player.

| Enemy | Observational | | | Interventional | | |
|---|---|---|---|---|---|---|
| | Slope | $p$-value | $r$ | Slope | $p$-value | $r$ |
| 1 | -0.008 | 0.08 | -0.055 | 0 | 0 | 0 |
| 2 | -0.029 | 4.17e−6 | -0.145 | 0 | 0 | 0 |
| 3 | -0.031 | 3.57e−4 | -0.113 | 0 | 0 | 0 |
| 4 | -0.023 | 5.97e−3 | -0.087 | 0 | 0 | 0 |
| 5 | -0.013 | 0.02 | -0.076 | 0 | 0 | 0 |

Table 8: Numeric results from regression analysis for the observational and interventional results in Figures 11b and c. The results indicate a very small strength of effect (slope) for both observational and interventional data and a small correlation coefficient ($r$), suggesting that there is, at best, only a very weak causal dependence of saliency on distance-to-player.

