# OpenReview forum: "Exploratory Not Explanatory: Counterfactual Analysis of Saliency Maps for Deep Reinforcement Learning"
_ICLR.cc/2020/Conference — Accept (Poster)_

### Official Review · AnonReviewer2 · 2019-10-21
**Official Blind Review #2**

**Rating:** 8

**Review:**

[score raised from weak accept to accept due to rebuttal/improvements]

Summary
The paper investigates the practice of using pixel-level saliency maps in deep RL to “explain” agent behavior in terms of semantics of the scene. The main problem, according to the paper, is that pixel-level saliency maps often correlate with semantic objects, however turning these correlations into explanations would require counterfactual analysis / interventions, which are almost never performed in practice. The paper highlights this issue with an extensive literature survey, and proposes a simple method to formulate “explanations” found via saliency maps into falsifiable hypotheses. Three experiments show how to apply the methodology in practice - in all three cases pixel-level correlations cannot be easily mapped to semantic-level explanations that hold (counterfactual) validation.

Contributions
The paper nicely summarizes the main contributions, namely: (i) a literature survey on pixel-saliency methods in deep RL and their use to “explain” agent behavior, (ii) a detailed description of the problem with the latter and a proposal to mitigate the main issues, and (iii) three experimental case-studies to illustrate the problem further and show how the proposed method can help.

Quality, Clarity, Novelty, Impact
The paper addresses a highly important issue in the field of interpretable deep learning. The main message is that a lack of scientific rigor, namely stating falsifiable hypotheses and validation of claimed hypotheses, can easily lead to misinterpretation of deep RL systems. This is a somewhat disenchanting message, but I personally think it is important to ensure that this message is heard in the field of interpretable ML in particular, and in the wider deep learning community in general. It is tempting to give simple answers to complex problems, and while I think saliency maps will play a large role in interpreting deep network decisions, I am also convinced that we need causal explanations, which salience maps (currently) cannot provide on a semantic level. The paper is well written and clear, the literature survey is quite extensive and valuable. The experimental results are nice, however they currently crucially lack quantitative statements that back up the qualitative results (see improvements below). While the latter must be included for publication, I am fairly confident that this can be rectified during the rebuttal phase and therefore (tentatively) vote for acceptance.

Improvements
a) Mandatory for publication! Back the results-plots up by numbers! In particular: visually estimating densities / correlations from scatter plots is often impossible and misleading - while the plots are nice to have, the claims regarding Figure 5, 8, 9 (b) and (c) must be backed up by reporting actual correlations / statistical tests. For instance, it is impossible to judge visually whether there’s any trend in 5 (c). Please report correlations for 5, 8, 9 (b) and perform suitable statistical tests for measuring increase/decrease in correlation for 5, 8, 9 (c).
Similarly, please report an appropriate metric to quantitatively judge the difference between the curves in 4, 6, 7 (c). It’s fine to include tables reporting the quantitative results in the appendix, they don’t necessarily have to be in the main paper.

b) Experimental details. Please report the details required to reproduce the experiments. In particular, what was the precise architecture for A2C and the hyper-parameter settings (particularly since the reference that is cited is not a paper, but a GitHub repo). For the figures, please report how saliency was measured exactly (was there a bounding-box around saliency/enemies? What was its size? Were intensities somehow normalized, were distances to enemies measured between centers of bounding-boxes, …?)

c) (Optional). It would be nice to see an example where the method is used but the original hypothesis is not rejected (i.e. there’s now stronger evidence for the original hypothesis due to the counterfactual analysis). I understand that this is beyond the scope of the rebuttal, and feel free to completely ignore this.


Major Comments
I) Please state whether the paper was written with feed-forward deep RL agents only in mind, or whether the paper is intended to also include recurrent deep RL agents (it would also be helpful to know whether the experiments used a feed-forward, or a recurrent version of A2C). While I think that many aspects carry over from feedforward architectures to recurrent ones, I personally think that some issues with counterfactual analysis could become more intricate with recurrent agents. For instance, on page 6, the described invariance in the first paragraph under ‘Counterfactual Evaluation of Claims’ is fine for feed-forward agents, but could be debatable with recurrent agents.  If you agree, please make this distinction clear in the paper (where appropriate) or state that the paper only applies to feedforward agents. If you disagree please indicate this during the rebuttal discussion.

II) Page 6, just above Sec. 5: “Since the learned policies should be semantically invariant under manipulations of the RL environment...”. I agree that they should ideally be invariant, for the semantic interventions to make sense, but please comment on whether this is a trivial assumption, how this assumption could (in principle) be verified and the potential consequences of this assumption being violated. I personally think that there’s a fair chance that the semantic space carved up by the agent (that potentially overfits a task/family of tasks) might be quite different from the semantic space given by the latent factors of the environment. This mismatch and its potential interference with the method should be discussed as a current shortcoming.

Minor Comments
I)  A potential subtlety (which I don’t expect you to resolve/discuss in the paper) is that feed-forward agents in an MDP environment can behave like recurrent agents by offloading memory into the environment. E.g. a breakout agent could “memorize” that it is in “tunnel-digging mode” by moving the paddle by a few pixels - this could then potentially shift it’s saliency away from the actual tunnel to the corresponding pixels around the paddle. Such cases might be very hard to interpret via saliency maps or interventional analysis, but I acknowledge that this is perhaps a more exotic case, given the current state of interpretable deep RL. Just a thought for future work perhaps...

**Experience Assessment:**

I have read many papers in this area.

**Review Assessment: Checking Correctness Of Derivations And Theory:**

I assessed the sensibility of the derivations and theory.

**Review Assessment: Checking Correctness Of Experiments:**

I assessed the sensibility of the experiments.

**Review Assessment: Thoroughness In Paper Reading:**

I read the paper at least twice and used my best judgement in assessing the paper.

---

> ### Comment · AnonReviewer2 · 2019-11-13
> **Waiting for the author's response**
>
> While there's still a bit of time left, I'd like to encourage the authors to engage in a discussion with the reviewers as early as possible, and certainly prepare a rebuttal and revised manuscript. Since the other two reviews are quite short and do not provide specific criticism or points to improve, perhaps the other reviewers haven't fully engaged with the paper and the main point did not come across yet.
>
> I personally think the paper is well written, has clear aims, and while I agree that there's no central one-line main-result, I think that the more prosaic style of the paper suits the topic really well. If the author's convincingly address improvements a) and b) and respond to the major comments, I am likely to raise my score.

---

> ### Author Response · Authors · 2019-11-14
> **Response to Review #2**
>
> We greatly appreciate Reviewer 2's extensive comments and suggested improvements. We have incorporated the suggested improvements to the best of our ability during the response period, and we summarize those revisions in a separate official comment. Specifically, we have provided statistics that quantify experimental effects, run and reported the results of hypothesis tests, and provided additional experimental details.  Below are some additional notes on specific comments from the review:
>
> Application to recurrent deep RL agents — As Reviewer 2 notes, the paper was written with feed-forward deep RL agents in mind. That said, the proposed methodology is post-hoc (i.e., not model-dependent), so aspects of the approach will carry over to recurrent RL agents. Our proposed methodology would not work for repeated interventions on recurrent RL agents due to their capacity for memorization. We have noted this distinction in Section 6.
>
> Semantic Invariance — We completely agree that the semantic space devised by the agent might be quite different from the semantic space given by the latent factors of the environment. It is crucial to note that this mismatch is one aspect of what plays out when researchers create hypotheses about agent behavior, and the methodology we provide in this work demonstrates how to evaluate hypotheses that reflect that mismatch.
>
> Positive Example — We agree it would have been ideal to provide an example where the original hypothesis was not rejected.  Unfortunately, we exhausted the set of obvious hypotheses for the two games we considered, and all were rejected.  We were surprised by these results, but they support the idea that saliency maps are easily misinterpreted.
>
> Memory in feed-forward agents — The subtlety that Reviewer 2 points about feed-forward agents behaving like recurrent agents by offloading memory into the environment is extremely interesting. Assessing whether memorization leads to different saliency behavior is a fascinating direction for future work.

---

> > ### Comment · AnonReviewer2 · 2019-11-14
> > **Thank you for the response and the updated manuscript**
> >
> > Thank you for addressing the points of improvement and major/minor comments raised in my review. I would have given the paper a higher score initially already, if the statistical analysis were present. It is crucially needed to verify the claims made based on the data in the paper. I am happy to see that the statistical analysis nicely confirms the results stated in the original manuscript. Together with the other improvements I am now confident to raise my score.

---

### Official Review · AnonReviewer1 · 2019-10-23
**Official Blind Review #1**

**Rating:** 3

**Review:**

Abstract:
The author suggests that saliency maps should be viewed as exploratory tools rather than explanation. The explore this idea in the context of a game.

Here is my main issue:
Although I believe there is a value in studies like this. I am not sure ICLR is the right venue for it. The paper is well written but it reads like a long opinion/blog-post. There is no overarching theory or generalizable observation not to mention a solution.
Yes, I agree that the method of interpreting the black box has a lot of issues and the counterfactual approach/causal approach is probably the right way to go but this is hardly news to the community.

In short: what the generalizable contribution of the paper?

I am open to change my mind if the discussion is convincing.

**Experience Assessment:**

I have read many papers in this area.

**Review Assessment: Checking Correctness Of Derivations And Theory:**

N/A

**Review Assessment: Checking Correctness Of Experiments:**

I did not assess the experiments.

**Review Assessment: Thoroughness In Paper Reading:**

I made a quick assessment of this paper.

---

> ### Author Response · Authors · 2019-11-14
> **Response to Review #1**
>
> We appreciate Reviewer 1’s comments, and address the main points of the review below:
>
> Generalizable contribution — The paper makes several contributions: (1) a survey of how saliency maps are currently used to explain the behavior of deep RL agents; (2) a new method to empirically evaluate the inferences made from saliency maps; and (3) an experimental evaluation that uses our proposed method to measure how well saliency maps correspond to the semantic-level inferences of humans.  Each of these contributions applies to any use of common saliency-map methods to understand the behavior of deep RL agents learned using feed-forward architectures.
>
> Overarching theory — As we describe in section 2, our proposed method is based on a formal theory of counterfactual intervention. Though the graphical model in Figure 2 represents an Atari game environment, researchers can reason about interventions in different vision-based RL domains by substituting different content for the state and pixels. We added a specific graphical model for Breakout in the Appendix (Figure 6) to clarify how the generalized causal graphical model in Figure 2 can be specified to a given domain. Section 6 contains additional points on the generalization of the proposed methodology.
>
> ICLR as a venue for this paper — ICLR is a nearly ideal venue for this work. The paper that first introduced saliency maps was published in a 2014 ICLR workshop (Simonyan et al. 2014). Subsequent ICLR papers have introduced new saliency map methods (e.g., Zintgraf et al. 2017) and analyzed these methods (e.g., Ancona et al. 2018). Many studies have critiqued the use of saliency maps in computer vision (Adebayo et al., 2018; Samek et al., 2018; Kindermans et al., 2019), but we are the first to analyze the utility  of saliency maps for understanding the behavior of deep RL agents. Finally, while methodological papers are relatively uncommon in machine learning conferences (including ICLR), effective evaluation of learned representations is vital to progress in the field. Saliency maps have become one of the primary methods to visualize the representations learned by deep neural networks, and better understanding the utility of saliency maps is central to understanding their proper role in research.
>
> Adebayo et al. "Sanity checks for saliency maps." NeurIPS 2018.
>
> Ancona et al. “Towards better understanding of gradient-based attribution methods for deep neural networks.” ICLR 2018.
>
> Kindermans et al. "The (un) reliability of saliency methods." Explainable AI: Interpreting, Explaining and Visualizing Deep Learning 2019.
>
> Samek et al. "Evaluating the visualization of what a deep neural network has learned." IEEE Transactions on Neural Networks and Learning Systems 2017.
>
> Simonyan et al. “Deep inside convolutional networks: visualising image classification models and saliency maps.” ICLR Workshop 2014.
>
> Zintgraf et al. “Visualizing deep neural network decisions: prediction difference analysis.” ICLR 2017.

---

> > ### Comment · AnonReviewer1 · 2019-11-14
> > **not a method but an exploratory analysis paper**
> >
> > I disagree with the authors that section 2 provides an overarching theory.
> >
> > First of all, the fact that there is a graphical model (which by the way was only used as schematic but not used for any inference of any kind) doesn't mean there is an overarching theory in the paper. Let me rephrase this: How does a causal graphical model (Fig 7 or Fig 2) helps/improves learning the agent and how do you incorporate it to the learning/inference of your method "algorithmically"?
> >
> > Second, as the author mentioned in page 3, using the saliency map to change the value of the pixel is not new. Other have done it: (Simonyan et al., 2014), (Zeiler & Fergus, 2014),  (Greydanus et al., 2017),  (Iyer et al., 2018).   --- so what is new here?
> >
> > Yes, the saliency map method was published in the ICLR, but the saliency map method is a general tool. Given a BlackBox f, it provides us with importance value for each feature/pixel/... We can deploy this tool on any blackbox. I am struggling with this paper: Given a game with an action set A and state set S and reward function R, (1) Are you proposing a new "Tool" to overcome the limitation of learning? By that I mean a tool that can be applied on any game or at least on a family of game? I don't see that. (2) you are providing new insight that we did not have about the saliency map in RL that we didn't have before?
> >
> > Having said that, I believe that the paper did a lot of experiments, and has a value an exploratory analysis paper (not a method paper). Perhaps, this what ICLR community wants. So if the AC rejects my vote, I am fine with it. I change my vote to weak reject.

---

> > > ### Author Response · Authors · 2019-11-15
> > > **Response to Reviewer #1**
> > >
> > > We appreciate the continued discussion with Reviewer 1 and the revised score. Below we address the additional questions raised in Reviewer 1’s latest comments:
> > >
> > > Goal of the paper — Reviewer 1 asks how the causal graphical model “improves learning the agent”?  Our proposed method is intended to improve *explanations* of deep RL agents rather than directly improve agent learning.  Better explanations do not directly improve learning, but they are vital for diagnosing errors in agent performance, forecasting how a given agent will perform in new environments, and improving the design of agents and agent training procedures.
> > >
> > > Value of the graphical model — Reviewer 1 is correct that the graphical model is used schematically to describe the methods we propose.  We provide the graphical model to be formal and clear about a key distinction between intervening on pixels and intervening on game state.  As we note below, such a distinction is the heart of this paper.
> > >
> > > Novelty of interventions — Reviewer 1 notes that many prior methods intervene on pixels and asks “what’s new here?”  Our primary contribution is to directly compare the inferences supported by intervening on pixels with the inferences supported by intervening on game state. Pixel-level interventions produce images for which the learned network function may not be well-defined and do not guarantee changes in semantic concepts or game state (see Table 1).  This distinction is shown clearly by the graphical model.  As we state in Section 4: “We generate counterfactual conditions by intervening on the RL environment.  Prior work has focused on manipulating the pixel input.  However, this does not modify the underlying latent game state. Instead, we intervene directly on game state.”
> > >
> > > Contribution — Reviewer 1 asks “are [you] providing new insight that we did not have about the saliency map in RL that we didn't have before?”  Yes, we provide a major new insight that saliency maps cannot be trusted as evidence for causal relationships between semantic concepts and agent behavior.  The strong evidence we provide to support this insight is a direct comparison between the inferences made by researchers using saliency maps and those that we can make using direct interventions on game state.

---

### Official Review · AnonReviewer3 · 2019-10-23
**Official Blind Review #3**

**Rating:** 1

**Review:**

The paper has a double aim. First, it is a survey on saliency maps used in explaining deep reinforcement learning models. Second, it is a proposal of a method that should overcome limitations of the current approaches described in the survey.
This double aim makes the paper hard to understand as the survey is not complete and the model is not well explained.
The main limitations the novel model aims to solve seems to be the production of "falsifiable" hypothesis in the explaination with saliency maps. However, experiments are really hard to follow and it is not clear why this is the case.

**Experience Assessment:**

I have read many papers in this area.

**Review Assessment: Checking Correctness Of Derivations And Theory:**

I did not assess the derivations or theory.

**Review Assessment: Checking Correctness Of Experiments:**

I assessed the sensibility of the experiments.

**Review Assessment: Thoroughness In Paper Reading:**

I read the paper at least twice and used my best judgement in assessing the paper.

---

> ### Author Response · Authors · 2019-11-14
> **Response to Review #3**
>
> We appreciate Reviewer 3’s comments.  The review has several main points that we address below:
>
> Double aim of the paper — Our principal contribution is a new method for empirical evaluation of explanations generated from saliency maps about the behavior of deep RL agents. We intentionally structured our paper to include both a survey of current practice and an application of our proposed approach.  Both elements were intended to aid reader understanding. The survey describes the inferences that require evaluation, and the application demonstrates the surprising conclusions supported by the evaluation method. We have attempted to improve our description of this approach (see “Clarity” below).
>
> Completeness of the survey — As we note in section 3, we surveyed 90 papers, each of which cited one or more key papers that described one of four saliency map methods.  We would be happy to include additional papers in our survey, expand our survey criteria, or consider additional updates to the survey, if Reviewer 3 could provide specific suggestions.
>
> Clarity — We have added additional details to Section 4 on how saliency is measured. We have also added more quantitative results from the experiments to Section 5 and Appendix E (see Tables 3, 6, 7 and 8), and more details on the model used for training in Section 5 and Appendix B. We hope these additions make the paper more clear, and we welcome additional recommendations.

---

### Author Response · Authors · 2019-11-14
**Summary of Revisions**

We made the following revisions to our posted paper:

Experimental Details:
- Described the architecture and hyper-parameters of the RL model (A2C) employed for the case studies (Section 5 and Appendix B).
- Described how saliency was measured in the case studies (Section 4 and 5).

Experiment Results:
- Added quantitative results for case studies 2 and 3 in Section 5 and the Appendix (Tables 3, 6, 7, and 8).
- Scaled saliency in Figure 5a to be more visible.
- Removed example plot in Figure 4a and instead added example saliency maps in Appendix A.
- Added a correlation plot to represent the relationship between saliency on score and agent behavior (Figure 4c).
- Added more description to captions on the Figures and Tables for clarity.

Discussion:
- Added a discussion on the applicability of the proposed methodology to recurrent deep RL agents (Section 6).
- Added an extended version of the causal graphical model in Figure 2 for Breakout to demonstrate generalizability (Figure 7 in Appendix D).

---

### Decision · Program_Chairs · 2019-12-19

**Decision:**

Accept (Poster)

**Comment:**

This was a contentious paper, with quite a large variance in the ratings, and ultimately a lack of consensus. After reading the paper myself, I found it to be a valuable synthesis of common usage of saliency maps and a critique of their improper interpretation. Further, the demonstration of more rigorous methods of evaluating agents based on salience maps using case studies is quite illustrative and compelling. I think we as a field can agree that we’d like to gain better understanding our deep RL models. This is not possible if we don’t have a good understanding of the analysis tools we’re using.

R2 rightly pointed out a need for quantitative justification for their results, in the form of statistical tests, which the authors were able to provide, leading the reviewer to revise their score to the highest value of 8. I thank them for instigating the discussion.

R1 continues to feel that the lack of a methodological contribution (in the form of improving learning within an agent) is a weakness. However, I don’t believe that all papers at deep learning conferences have to have the goal of empirically “learning better” on some benchmark task or dataset, and that there’s room at ICLR for more analysis papers. Indeed, it’d be nice to see more papers like this.

For this reason, I’m inclined to recommend accept for this paper. However this paper does have weaknesses, in that the framework proposed could be made more rigorous and formal. Currently it seems rather adhoc and on a task-by-task basis (ie we need to have access to game states or define them ourselves for the task). It’s also disappointing that it doesn’t work for recurrent agents, which limits its applicability for analyzing current SOTA deep RL agents. I wonder if authors can comment on possible extensions that would allow for this.